# Comparative Analysis of *Tylosema esculentum* Mitochondrial DNA Revealed Two Distinct Genome Structures

**DOI:** 10.3390/biology12091244

**Published:** 2023-09-16

**Authors:** Jin Li, Christopher Cullis

**Affiliations:** Department of Biology, Case Western Reserve University, Cleveland, OH 44106, USA; jxl1269@case.edu

**Keywords:** legume mitogenome, mitogenome assembly, PacBio HiFi sequencing, mitogenome diversity, genome structural variation, heteroplasmy, horizontal gene transfer, phylogeny

## Abstract

**Simple Summary:**

*Tylosema esculentum* is a potential crop in southern Africa, known for its high nutritional value and ability to thrive in long-term drought conditions. To gain a better understanding of its genetic characteristics, the mitogenomes of 84 individuals from different locations in Namibia and South Africa were assembled and compared using both PacBio and Illumina sequencing data. The study revealed two distinct germplasms with significant differences in the mitogenome structure and sequence. Comparative genomics analysis was conducted to explore heteroplasmy and horizontal transfer in the marama mitogenome, providing valuable insights into cytoplasmic genetic diversity and inheritance. Additionally, evolutionary studies were performed on marama and its related legumes, indicating that the Cercidoideae subfamily tends to possess a more complete set of mitochondrial genes compared to the Faboideae species. The findings made in this study not only contribute to the germplasm selection and future marama breeding but also enhance our understanding of the inheritance and evolution of plant mitogenomes.

**Abstract:**

*Tylosema esculentum*, commonly known as the marama bean, is an underutilized legume with nutritious seeds, holding potential to enhance food security in southern Africa due to its resilience to prolonged drought and heat. To promote the selection of this agronomically valuable germplasm, this study assembled and compared the mitogenomes of 84 marama individuals, identifying variations in genome structure, single-nucleotide polymorphisms (SNPs), insertions/deletions (indels), heteroplasmy, and horizontal transfer. Two distinct germplasms were identified, and a novel mitogenome structure consisting of three circular molecules and one long linear chromosome was discovered. The structural variation led to an increased copy number of specific genes, *nad5*, *nad9*, *rrnS*, *rrn5*, *trnC*, and *trnfM*. The two mitogenomes also exhibited differences at 230 loci, with only one notable nonsynonymous substitution in the *matR* gene. Heteroplasmy was concentrated at certain loci on chromosome LS1 (OK638188). Moreover, the marama mitogenome contained an over 9 kb insertion of cpDNA, originating from chloroplast genomes, but had accumulated mutations and lost gene functionality. The evolutionary and comparative genomics analysis indicated that mitogenome divergence in marama might not be solely constrained by geographical factors. Additionally, marama, as a member from the Cercidoideae subfamily, tends to possess a more complete set of mitochondrial genes than Faboideae legumes.

## 1. Introduction

*Tylosema esculentum*, also known as the marama bean and the gemsbok bean, is a non-nodulated legume from the Fabaceae family and the Cercidoideae subfamily [1,2]. Marama is endemic to the Kalahari Desert and the surrounding arid and semi-arid regions of Botswana, Namibia, and South Africa [3]. Marama has developed a special drought avoidance strategy by growing giant tubers that can weigh more than 500 pounds to store water, helping them survive the harsh conditions of long-term drought and little rainfall [4]. The seeds of marama are edible and nutritious, with a protein content of approximately 30–39% dry matter (dm) and a lipid content of 35–48% dm, comparable to those found in commercial crops such as soybean and peanuts, respectively [5,6,7]. Marama also contains many micronutrients and phytochemicals that are beneficial to human health [8,9,10]. However, marama is still mainly collected from wild plants, and the domestication of marama has long been thought to have the potential to improve local food security. Marama usually does not flower and produce seeds until two to four years after planting, making traditional breeding very inefficient [11]. Therefore, selecting improved marama individuals and exploring the underlying genetic diversity is considered of great significance for the improvement of the bean [12].

Mitochondria in eukaryotic cells are generally considered to have originated from the endosymbiosis of alpha-proteobacteria, although a number of changes have occurred since then, including the loss of many genes and their transfer to the nuclear genome [13]. The mitochondrial genomes of animals and plants have been found to vary greatly. Animal mitogenomes are usually small, only about 16 kb in size, and contain 37 genes [14]. Plant mitogenomes are commonly larger, with 50 to 60 genes and expanded intergenic non-coding regions that result from the DNA transfer from other cellular compartments or even from different organisms [15,16,17]. In land plants, they range in size from 66 kb in *Viscum scurruloideum* to 11.3 Mb in *Silene conica* [18,19].

Plant mitochondrial genes are very conserved, are considered to play important roles in ATP synthesis, and are also related to plant fertility and environmental adaptation [20,21,22]. The base substitution rate of mitochondrial genes is lower than that of chloroplast genes, and it is far lower, only about one-tenth of the rate, when observed in nuclear genes [23,24]. Plant mitogenomes evolve even up to 100 times slower than animal mitogenomes [25]. However, the structure of plant mitochondrial genomes is very dynamic, with a large number of sequence rearrangements, and repeat-mediated homologous recombination plays an important role in its structural variations [26,27].

In the mitochondrial genomes of many angiosperms, repetitive sequences account for 5–10% of the total genome size, and in a few plants such as *S. conica* and *Nymphaea colorata*, the proportion can exceed 40% with repetitive fragments up to 80 kb in length [18,28]. Recombination mediated via short or intermediate length repeats is considered to be less frequent, but recombination on long repeats (>1 kb) is thought to occur more frequently and usually generates equimolar-recombined molecules in the plant mitogenomes [29,30,31].

The third-generation sequencing technology, such as PacBio, provides long reads with an average length of 10–25 kb, spanning the long repeats in the plant mitogenome. This makes the study of structural variations caused by the long repeat-mediated recombination possible [32,33]. High sequencing coverage is also important, not only to make the genome assembly more reliable, but also to determine the proportion of different chromosome structures. It is also indispensable for the accurate assessment of nucleotide polymorphisms [34]. The latest PacBio HiFi sequencing with an extremely high accuracy of 99.9%, which needs little correction via the data generated from other sequencing platforms, further promotes the genome assembly [35,36].

Although plant mitochondrial genomes are often reported as one master circular chromosome, in reality, they often exist as multipartite structures. This includes a combination of linear molecules, branched molecules, and subgenomic circular molecules [37,38]. For example, the mitogenome assembly of *Solanum tuberosum* was found to contain at least three autonomous chromosomes, including two small circular molecules and a long linear chromosome of 312,491 bp in length [39].

The genome of *T. esculentum* is small, with an estimated haplotype genome size of only 277.4 Mb [40]. Previous studies of its mitogenome found that two different equimolar structures coexist in the same individual: two autonomous rings with a total length of 399,572 bp, or five smaller circular molecules [31]. These two structures are believed to be interchangeable via recombination on three pairs of long direct repeats (3–5 kb in length). As described in the study of the *Brassica campestris* mitogenome, recombination on a pair of 2 kb repeats was postulated to split the 218 kb master chromosome into two subgenomic circular molecules of 135 kb and 83 kb in length [41].

The previous comparative analysis of 84 *T. esculentum* chloroplast genomes has found two distinct germplasms. The two types of chloroplast genomes are different from each other at 122 loci and at a 230 bp inversion [42]. Among many of these loci, heteroplasmy, the existence of two or more different alleles, could be seen, albeit one generally at a frequency below 2%. The occasional paternal leakage is considered to cause this phenomenon [43,44]. The reason for its stable inheritance is thought to be related to the developmental genetic bottleneck, but the specific mechanism is still unclear [45].

Research on heteroplasmy should avoid interference from homologous sequences of mitochondrial DNA in other organelles. In fact, the horizontal gene transfers of DNA from the chloroplast genome to the mitogenome and the nuclear genome, or between the mitogenome and the nuclear genome, are very common [46,47]. The transfer of DNA from the mitogenome to the chloroplast genome is very rare, but it has been reported in a few studies [48,49]. Many chloroplast genes have been found to become pseudogenes and lost function after being inserted into the mitogenome, but the reason behind it is still unclear [50].

Although the mechanisms underlying the effects of cytoplasmic activities on agronomic traits are not well understood, previous studies on potato cytoplasmic diversity have found that cytoplasmic types are directly related to traits such as tuber yield, tuber starch content, disease resistance, and cytoplasmic male sterility [51,52,53]. Furthermore, certain mtDNA types and certain chloroplast DNA types were found to be linked [54]. A comparative genomics analysis based on the chloroplast genomes of 3018 modern domesticated rice cultivars found that their genotypes fall into two distinct clades, suggesting that the domestication of these cultivars may have followed two distinct evolutionary paths [55]. These studies have the potential to reveal important selections occurring in organelle genomes, improving the understanding of plant adaptation to different environments and providing a basis for crop breeding to increase the yield in the corresponding environments.

The chloroplast genome is widely used in research on plant evolution, but the comparative analysis based on the plant mitochondrial genome is not so extensive, and there is even less research on the mitochondrial diversity within the same species [56,57]. In this study, the diversity of the marama mitochondrial genome was analyzed by mapping the WGS reads of 84 *T. esculentum* individuals to the previously assembled reference mitogenome aiming to (1) discover possible mitogenome structural diversity and the impact of structural variations on the gene sequence and copy numbers; (2) compare the differential loci in the mitogenomes of 43 independent marama individuals collected from different geographical locations in Namibia and South Africa to explore the divergences that have occurred and the possible decisive environmental factors behind them; (3) look at heteroplasmy, the co-existence of multiple types of mitogenomes within the same individuals, and compare the allele frequencies in related individuals to better understand the underlying cytoplasmic inheritance; and (4) track polymorphisms accumulated in the chloroplast DNA insertion and interpret the fate of the inserted gene residues. In addition, the conserved protein-coding genes from the mitogenomes of *T. esculentum* and other Fabaceae species were compared to explore the evolutionary relationship between them.

## 2. Materials and Methods

### 2.1. Plant Materials and DNA Extraction

Samples 4 and 32 were two individuals in the greenhouse of Case Western Reserve University grown from seeds collected in Namibia at (22°1′33″ S 19°8′10″ E) and (22°19′33″ S 19°38′7″ E), respectively. They were identified via PCR amplification to have type 2 and type 1 germplasms, respectively. One gram of fresh young leaves was collected from the two plants and ground thoroughly with a pestle in a mortar containing liquid nitrogen. The DNA was then extracted using a Quick-DNA HMW MagBead kit (Zymo Research) following the protocol. The double-stranded DNA was quantified via the InvitrogenTM QubitTM 3.0 Fluorometer after mixing 5 μL of DNA with 195 μL of the working solution, and 200 ng of DNA was electrophoresed on a 1.5% agarose TBE gel at 40 V for 24 h. The plant materials included an additional 131 individuals, 84 of which had WGS sequencing data available and were stored in the NCBI SRA database (PRJNA779273), as described in the previous study (Appendix A) [42]. Forty of these plants were progeny grown from the seeds of seven accessions collected from different geographical locations in Namibia for the study of cytoplasmic inheritance. The remaining samples were collected from wild plants in different geographical regions and considered as independent individuals for the study of cytoplasmic genomic differences and evolution. This collection includes new plants not introduced in previous studies: 8 from Tsjaka (22°47′15.5″ S 19°12′35.8″ E), 8 from Otjiwarongo (20°27′49.3″ S 16°38′51.8″ E), 18 from D1776 (22°19′44.3″ S 19°38′04.5″ E), and 13 from the Namibia Farm (21°23′48.5″ S 19°44′59.6″ E).

### 2.2. High-Throughput Sequencing

The DNA extracted from Samples 4 and 32 (10 micrograms or more per sample) was sent to the Genomics Core Facility at the Icahn School of Medicine at Mount Sinai for sequencing. The HiFi sequencing libraries were prepared using the SMRTbell^®^ express template prep kit 2.0 (Pacific Biosciences, Menlo Park, CA, USA). SMRT sequencing was performed on four 8M SMRT^®^ Cells (two per sample) on the Sequel^®^ II system. An amount of 2,184,632 PacBio HiFi reads with a total length of 21.6 G bases were generated for Sample 4 and 498 Mb reads for Sample 32. In addition, whole-genome sequencing of DNA from fresh young leaves or the embryonic axis of germinating seeds from the 84 samples collected in Namibia and South Africa was performed using the Illumina platform, as described in the previous study [42].

### 2.3. Mitogenome Assembly and Annotation

The PacBio HiFi long reads were assembled using the HiCanu assembler (https://canu.readthedocs.io/en/latest/quick-start.html#assembling-pacbio-hifi-with-hicanu, accessed on 15 October 2022) [58]. The input genome size was set to 2 Mb to obtain more complete organelle genome contigs. The PacBio long reads spanning the ends of the contigs were used to further scaffold the assembly. The assembled mitogenome was annotated using MITOFY (https://dogma.ccbb.utexas.edu/mitofy/, accessed on 5 November 2022), BLAST (https://blast.ncbi.nlm.nih.gov/Blast.cgi, accessed on 5 November 2022) [59,60], and tRNAscan-se 2.0 (http://lowelab.ucsc.edu/tRNAscan-SE/, accessed on 10 November 2022) [61]. The assembly was verified by mapping the Illumina reads and PacBio HiFi reads from different individuals to the generated mitogenome sequences using Bowtie 2 v2.4.4 (https://github.com/BenLangmead/bowtie2, accessed on 20 November 2022) [62] and pbmm2 v1.10.0 (https://github.com/PacificBiosciences/pbmm2, accessed on 25 November 2022) [63], respectively. The alignment was visualized and checked in IGV (https://software.broadinstitute.org/software/igv/, accessed on 25 November 2022) [64], showing no ambiguity. The mitochondrial gene arrangement maps of *T. esculentum* were generated via the OGDRAW software tool v1.3.1 (https://chlorobox.mpimp-golm.mpg.de/OGDraw.html, accessed on 10 December 2022) [65].

### 2.4. Mitogenome Polymorphism

The program NUCmer in MUMmer4 (https://mummer4.github.io/index.html, accessed on 5 November 2022) [66] was used to locate highly conserved regions in the two types of mitochondrial genomes of *T. esculentum*. The alignment was visualized via a synteny diagram drawn using the RIdeogram package in R (https://github.com/TickingClock1992/RIdeogram, accessed on 15 November 2022) [67].

The 2108 bp type 2 unique fragment was blasted in the NCBI database for potential origin. NCBI Primer-BLAST (https://www.ncbi.nlm.nih.gov/tools/primer-blast/, accessed on 10 November 2022) was utilized to design two pairs of primers [68], to amplify across the two ends of this fragment via PCR to confirm its presence. The primers to amplify across the left end included the left forward primer (GAGACCGAGCGCAAGAACTA) and the left reverse primer (TCAGATGGCTAAACAGGCGG), and the product size was 990 bp. The primers to amplify across the right end included the right forward primer (CGCTCGTGACTCATTGAGGA) and the right reverse primer (TTGGTAAGCGGATGCTCTGG), and the product size was 289 bp.

An amount of 20 uL of mixtures were prepared by separately mixing the DNA from five randomly selected *T. esculentum* samples and the GoTaq Master Mix (Promega, Madison, WI, USA). The amplifications started with denaturation at 95 °C for 5 min, followed by 30 cycles of 95 °C for 45 s, 54 °C for 45 s, and 72 °C for 1 min, and a final 72 °C for 5 min.

The Illumina reads from the 84 individuals were mapped to the type 1 reference mitogenome of *T. esculentum* (OK638188 and OK638189) using Bowtie 2 v2.4.4. SNPs, indels were searched using SAMtools 1.7 mpileup (http://www.htslib.org/, accessed on 5 December 2022) and BCFtools 1.8 call (http://www.htslib.org/doc/1.8/bcftools.html, accessed on 5 December 2022) [69], and the alignment was visualized in IGV to find heteroplasmy manually. To minimize the interference of sequencing errors and strand bias, only alleles with a frequency of at least 2%, a Phred score above 20, and presence in the strands in both orientations were recorded [42]. Reads in the nuclear genome and the plastome homologous to the mtDNA were excluded. The alleles from the mitochondrial or chloroplast genomes were distinguished via their frequency at the differential loci on the 9798 bp chloroplast DNA insertion.

### 2.5. Mitogenome Divergence

A pairwise comparison was performed on the divergent mitogenomes of *T. esculentum* (OK638188 and OK638189) and six other Fabaceae species, including *Cercis canadensis* (MN017226.1), *Lotus japonicus* (NC_016743.2), *Medicago sativa* (ON782580.1), *Millettia pinnata* (NC_016742.1), *Glycine max* (NC_020455.1), and *Vigna radiata* (NC_015121.1) using the program PROmer [70] in MUMmer4 to detect the syntenic regions. The alignments were then visualized via the Rideogram package in R. A synteny block diagram was drawn via Mauve 2.4.0 (https://darlinglab.org/mauve/mauve.html, accessed on 15 December 2022) [71] and the genes contained in each block were marked on the plot.

### 2.6. SSR Analyses

Microsatellites were analyzed via MISA (https://webblast.ipk-gatersleben.de/misa, accessed on 20 December 2022) [72] on the type 1 reference mitogenome of *T. esculentum*, looking for repeats of motifs one to six base pairs long. The minimum number of repetitions were set to 10, 6, 5, 5, 5, and 5 for mono-, di-, tri-, tetra-, penta-, and hexanucleotide repeats, respectively.

### 2.7. Phylogenetic Tree Construction

Two phylogenetic trees were constructed separately: one based on the mitogenome conserved gene sequences to explore the evolutionary relationship between *T. esculentum* and the other selected legumes, and the other tree was built on all differential loci found in the mitogenomes of *T. esculentum* to explore the inter-population and intra-population relationship among the 43 independent samples.

The 24 conserved mitochondrial genes, *atp1*, *atp4*, *atp6*, *atp8*, *atp9*, *nad3*, *nad4*, *nad4L*, *nad6*, *nad7*, *nad9*, *mttB*, *matR*, *cox1*, *cox3*, *cob*, *ccmFn*, *ccmFc*, *ccmC*, *ccmB*, *rps3*, *rps4*, *rps12*, and *rpl16* from the mitogenomes of *T. esculentum* (OK638188 and OK638189), *Arabidopsis thaliana* (NC_037304.1), *C. canadensis* (MN017226.1), *L. japonicus* (NC_016743.2), *M. sativa* (ON782580.1), *M. pinnata* (NC_016742.1), *G. max* (NC_020455.1), and *V. radiata* (NC_015121.1) were concatenated to make artificial chromosomes. A Maximum Likelihood (ML) phylogenetic tree using the Jukes–Cantor model was built in Mega 11 (https://www.megasoftware.net/, accessed on 15 January 2023) [73] after the chromosomes were aligned via Muscle v5 (https://www.drive5.com/muscle/, accessed on 10 January 2023) [74]. The topology was validated using a Bayesian inference phylogenetic tree drawn via BEAST v1.10.4 (https://beast.community/, accessed on 25 January 2023) [75] and FigTree v1.4.4 (http://tree.bio.ed.ac.uk/software/figtree/, accessed on 25 January 2023).

Artificial chromosomes concatenated via 40 bp segments at each of the 254 differential loci found in the mitogenomes of *T. esculentum* were prepared for the 43 independent individuals and aligned via the Muscle v5. A Maximum Likelihood (ML) phylogenetic tree using the Jukes–Cantor model was drawn on the 43 chromosomes. Frequencies from 1000 bootstrap replicates were labeled on the branches with 60% as the cutoff. The topology was verified via a neighbor-joining tree in Mega 11.

### 2.8. Genetic Information Exchange between Organelles

The reference chloroplast genome sequence of *T. esculentum* (KX792933.1) was blasted to its type 1 reference mitogenome (OK638188 and OK638189) and visualized via the Advanced Circos function in Tbtools (https://github.com/CJ-Chen/TBtools, accessed on 25 January 2023) [76,77]. Primers were designed to verify the presence of the 9798 bp long homologous fragment in both the mitochondrial and chloroplast genomes. Four pairs of primers were designed using NCBI Primer-BLAST to amplify the products spanning the two ends of the 9798 bp cpDNA insertion in the mitochondrial and chloroplast genomes, respectively. The two pairs of primers designed based on the mitogenome sequence included MitoLL Forward (ACGCAGAAAAGAGGCCGAA) and MitoLR Reverse (CCTTCGTTTAAGAGAATGTTTTTGG), with a product size of 117 bp, and MitoRL Forward (TCTTTGCTACAGCTGATAAAAATCG) and MitoRR Reverse (CCTATGTTCGTTTTCGCCCTG), with a product size of 120 bp. The two pairs of primers designed based on the plastome sequence included ChlLL Forward (CGTAGTCGGTCTGGCCC) and MitoLR Reverse (CCTTCGTTTAAGAGAATGTTTTTGG), with a product size of 117 bp, and MitoRL Forward (TCTTTGCTACAGCTGATAAAAATCG) and ChlRR Reverse (GCTTTTAATAATATGGCCGTGATCT), with a product size of 120 bp.

An amount of 20 µL of mixtures were prepared by separately mixing the DNA from two randomly selected *T. esculentum* samples and the GoTaq Master Mix (Promega, Madison, WI, USA). The amplifications started with denaturation at 95 °C for 5 min, followed by 32 cycles of 95 °C for 30 s, 55 °C for 30 s, and 72 °C for 30 s, and a final 72 °C for 5 min.

## 3. Results

### 3.1. Genome Structure and Rearrangement

When the WGS Illumina reads from the 84 individuals were mapped to the two chromosomes of the reference mitogenome of *T. esculentum*, LS1 (OK638188) and LS2 (OK638189), two distinct mitogenomes were found. The mitogenomes of 45 individuals were similar to the reference mitogenome, with only a few substitutions and indels seen, termed as type 1. However, the mitogenomes of the other 39 individuals were similar to each other but differed substantially from the reference, including both structural and sequence differences, termed as type 2 (Appendix A). This is consistent with the previously published study of *T. esculentum* chloroplast genomes, where these 84 individuals were found to contain two distinct germplasms [42]. These two cytotypes actually differ not only in the chloroplast genome but also in the mitochondrial genome, but no individuals with a type 1 chloroplast genome and a type 2 mitogenome, or vice versa, have been identified. The PacBio HiFi reads from the type 2 individual Sample 4 were assembled via Canu to generate three circular molecules M1 (OP795449), M2 (OP795450), and M4 (OP795447), and one linear chromosome M3 (OP795448), with a total length of 436,568 bp and a GC content of 44.8% (Table 1 and Appendix A). The four chromosomes consisted of 21 contigs assembled directly from the Illumina reads of type 2 individuals, containing four double-copy regions and one triple-copy region (Figure 1 and Table 2). Among these multi-copy regions, homologous sequences of contigs H and I were also doubled in coverage in the type 1 mitogenome of *T. esculentum*, but the rest were present as single-copy sequences in the type 1 mitogenome [31].

Both ends of chromosome M3 were found to be long repeats that were homologous to parts of other chromosomes (Figure 2). In addition, in a very long range of 15 to 20 kb, the sequencing depth gradually decreased towards both ends with many PacBio reads reaching the ends within this range. This furthers confirms that this is a linear chromosome that exists in different lengths in cells because of the lack of telomere protection. Linear chromosomes have been found to stably exist in eukaryotic cells even in the absence of telomeres via strand invasion between terminal sequences and their homologous internal sequences to form t-loops to protect the chromosomes from degradation [78]. Since the repeats at both ends of M3 are very long, the PacBio reads we obtained cannot span them to verify whether this linear molecule recombines with other chromosomes. Long range PCR amplification that can amplify sequences above 20 kb can be considered here to answer this question.

Numerous differences were found between the type 1 and type 2 mitogenome structures (Figure 3). One is a rare recombination on a pair of *trnfM* genes on the large circular molecule M1, which inverts a 12,846 bp sequence. The sequence before inversion also appeared in type 2 plants, but with a frequency less than 2%. Furthermore, the three small rings of the type 1 mitogenome exist in different forms in the type 2 mitogenome, and four gaps have been found on them. New type 2 exclusive fragments were discovered, including a 2108 bp segment, which connected the originally distantly located contigs C1 and B. In addition, the recombination on a pair of 35 bp direct repeats joined contigs C1 and D2 to form a new circular molecule M2. C2 was found to connect with K1 and then further extended to N and L2 to form a linear chromosome, but the mechanism behind this is unclear. It can be seen that the deletion and insertion of the entire DNA segment, alongside repeat mediated recombination, can lead to dramatic changes in the mitogenome structure.

The two types of *T. esculentum* mitogenomes were compared via the NUCmer alignment and then visualized via a synteny diagram, showing a high degree of similarity (Figure 4). The basic blocks making up the two mitogenomes are highly similar, except for the 2108 bp type 2 unique fragment and some other short insertions and deletions, but the order of these blocks has been changed via recombination. When the WGS Illumina reads from the 84 individuals were mapped to the region where the 2108 bp type 2 unique fragment resides, 39 samples were found to contain this fragment while 45 plants did not (Appendix A). Furthermore, in the type 1 mitogenome, LS1 and LS2 are two autonomous circular chromosomes that do not recombine into one master circle, but in the type 2 mitogenome, a linear chromosome M3 was found to contain homologous sequences from both LS1 and LS2, suggesting that these molecules may not be independent of each other in all marama individuals.

Blast results showed that this 2108 bp type 2 mitogenome exclusive fragment was highly similar to the mitochondrial sequences of the Fabaceae species, *Lupinus albus* and *Indigofera tinctoria*, suggesting that they share a common ancestor (Figure 5A). Two pairs of primers were designed and found to effectively identify this 2108 bp fragment (Figure 5B). As shown in Figure 5C, samples A and B, out of the five randomly selected samples, contained both the 990 bp left end band and the 289 bp right end band after amplification, indicating that only these two of the five had type 2 mitogenomes.

### 3.2. Gene Annotation

Both types 1 and 2 mitochondrial genomes of *T. esculentum* share identical sets of genes but differ in the copy numbers of certain genes. These genomes contain 35 unique protein-coding genes, 3 unique rRNA genes, and 16 distinct tRNA genes (Figure 6; Table 3) [31]. The type 2 mitogenomes have two copies of *nad9* and *atp8*. The gene *nad9* is located on contig K1, a long repeat present on both chromosomes M3 and M4 in type 2 mitogenomes, while there is only one copy of K1 in type 1 mitogenomes. The gene *atp8* is located on a pair of long repeats J, so its copy number is doubled in both types of mitogenomes. The copy number of exon 3 and 4 of gene *nad5* is also doubled in type 2 mitogenomes but not in type 1, and it is not known whether this affects its expression level. In addition, there are two copies of *rrn5* and *rrnS* in type 2 mitogenomes but only one copy in type 1. A total of 26 tRNA genes were found in type 2 mitogenomes, including four copies of *trnfM-CAT*, three copies of *trnM-CAT*, three copies of *trnC-GCA*, two copies of *trnP* and *trnQ*, and 12 single-copy tRNA genes.

The atypical start codon ACG was used by three genes *nad1*, *nad4L*, and *rps10*, and ATT was used by the gene *mttB*. This is consistent with the research on the mitogenome of common beans from which these four genes were all reported to use an alternative initiation codon ACG [79]. C-to-U editing was found to be widely used in mitochondrial and chloroplast genes in land plants [80]. ATT is also usually used as an alternative start codon in the mitogenome. For example, *mttB* in *Salix purpurea* was reported to use an ATT [81].

### 3.3. Mitogenome Divergence

The mitogenome of *T. esculentum* was highly divergent from those of the six selected legume species, which covered 23% to 48% of the marama mitogenome, ranging in length from 91.9–191.8 kb (Figure 7). Of these species, *C. canadensis* was most closely related to marama, while *M. sativa* was the least similar to marama. The mitogenomes of *G. max*, *L. japonicus*, and *V. radiata* all contain homologous sequences covering 25% of the marama mitogenome, equal to 99.89 kb in length. *Bauhinia variegata* is closer to marama than *C. canadensis* in the phylogenetic tree, but its mitogenome sequence is not available in NCBI GenBank [82].

The loss of mitochondrial protein-coding genes and the functional transfer of these genes from the organelle genome to the nuclear genome are common in the evolution of angiosperms, but some plants tend to retain a more complete set of mitochondrial genes [83]. The mitogenomes of the Cercidoideae species, *T. esculentum* and *C. canadensis* contain functional protein-coding genes *sdh3*, *sdh4*, and *rpl10*, which have been lost in many other legumes (Table 4). Rare gene losses were also seen in the mitogenomes of some legumes, such as *rpl5* in *G. max*, *cox2* in *V. radiata*, and *rps1* in *L. japonicus*, but these genes all remain intact and functional in *T. esculentum* and *C. canadensis* [84,85,86].

In a pairwise comparison of the mitogenomes of seven legume species, the synteny plot revealed numerous rearrangements and a high degree of divergence among the mitogenomes of even closely related species (Figure 8). The mitogenomes of *C. canadensis* and *T. esculentum* contain many distinct regions but also some long homologous segments.

The phylogenetic tree shown in Figure 9 was built on the 24 conserved mitochondrial protein-coding genes *atp1*, *atp4*, *atp6*, *atp8*, *atp9*, *nad3*, *nad4*, *nad4L*, *nad6*, *nad7*, *nad9*, *mttB*, *matR*, *cox1*, *cox3*, cob, *ccmFn*, *ccmFc*, *ccmC*, *ccmB*, *rps3*, *rps4*, *rps12*, and *rpl16*, which are present in all these eight species. This tree is consistent with previously published phylogenetic trees constructed on chloroplast protein-coding genes [87,88]. As another species of Cercidoideae, *C. canadensis* was expected to be the closest relative in these plants to *T. esculentum*. Among the Faboideae species, *M. sativa* and *L. japonicus* are closely related, and *V. radiata*, *G. max*, and *M. pinnata* belong to another clade.

### 3.4. Nucleotide Polymorphism

Seventeen haplotypes were found in these 47 plants, which could be clearly divided into two groups: namely, the type 2 plants from the Namibian farm and the UP farm, and the remaining type 1 plants (Figure 10). The mitogenomes of type 2 plants are relative conserved, which may be caused by sampling errors, and a larger sample size is needed to verify. The only differences between type 2 plants were four deletions at four closely located loci on chromosome LS2. However, the type 1 plants can be divided into many groups based on the variations. Some geographical patterns can be seen in the distribution of variation. For example, in the four plants from Osire, there were three substitutions on chromosome LS1, A > C at 146,140 bp, C > A at 173,053 bp, and C > A at 217,508 bp, and a deletion at 128,192 bp on chromosome LS2. These are variations unique to Osire plants. Similar geographic-specific variation can be seen in plants elsewhere. However, our data may still have sampling issues. For example, only a single sample was collected in some regions including Otjiwarongo and Okamatapati. In addition, although distant wild individuals in each geographic region were intentionally selected, there is no guarantee that they are not related. The findings here still need to be validated by sequencing more samples and studied alongside the phenotypic performance of the plants to determine whether any of these variations are the result of plants evolving to better adapt to different environments.

Type 2 refers to the seven Index plants from the Pretoria Farm excluding Index8 (initially collected from Namibia but the exact location is unknown), 29 M of descendent plants originally from the Namibia Farm excluding M40, and two individuals R1R2 and nar16 of unknown origin. Type 1 represents all remaining plants, including A plants, S plants, Index8, and M40. Numbers in parentheses indicate the counts of exclusive variants of this type. LS1 and LS2 are type 1 marama reference mitochondrial chromosomes in GenBank with accession numbers OK638188 and OK638189.

A total of 254 differential loci were found in the mitogenomes of the 84 *T. esculentum* individuals, including 143 SNPs, 52 insertions, and 59 deletions (Table 5). Type 1 and type 2 mitogenomes differed at 230 loci, including 129 substitutions, 50 deletions, and 51 insertions. The mitogenomes of type 2 plants differed at only 4 loci, whereas that of type 1 plants differed at 24 loci.

The mitochondrial gene sequence of *T. esculentum* is very conserved. A total of 11 variations were found in the mitochondrial gene sequence, and only one of them was in the coding sequence, which was a 2368 A > G substitution, which resulted in a N303D change in the gene *matR* (Table 6). Furthermore, 10 of the 11 variations were found on one subgenomic ring LS2 of the reference mitogenome of *T. esculentum*. Whether chromosome LS1 is more conserved than chromosome LS2 is unknown. Although the intergenic spacer of LS1 contained more variations than LS2, the gene sequence on LS1 appeared to be more conserved, and the cpDNA insertions were also found rarely in LS1 but abundantly in chromosome LS2.

In the phylogenetic tree constructed on the differential loci of the mitogenomes of *T. esculentum*, the two types of germplasms fell into two clusters, as expected (Figure 11). However, using PCR amplification on the newly collected samples, it was found that there may not be obvious differences in geographical distribution between type 1 and type 2 plants. Both types were detected in Otjiwarongo and the Namibia Farm. Among them, three out of eight plants collected in Otjiwarongo were identified as containing the type 2-specific fragment, and 2 out of 13 plants from the Namibian Farm were identified as type 2 individuals.

Furthermore, the type 1 plants were then divided into groups. Plants from Epukiro, Osire, Ombujondjou, and Otjiwarongo belonged to one clade, and they all had GCC at positions 127,684 to 127,686 on chromosome LS1 (OK638188). The remaining plants all had the alternative alleles TAA. However, Sanger sequencing of the newly collected samples found that two out of eight individuals from Otjiwarongo also contained TAA, eight plants from Tsjaka instead contained GCC, 5 out of 13 plants from the Namibian Farm contained TAA, and the remaining eight contained GCC. Thus, this widespread difference in marama mitogenomes may not be geographically restricted either. In addition, despite Tsumkwe and Tsjaka being two distant sampling sites, the plants from these locations were clustered in one clade, as also observed in the phylogenic tree constructed using the complete chloroplast genome. This raises questions about whether there are factors other than geographical distance that are determining the grouping of these plants.

### 3.5. SSRs and Heteroplasmy Analyses

A total of 48 SSR motifs were found via MISA in the reference mitogenome of *T. esculentum*, LS1 (OK638188) and LS2 (OK638189), of which 38 were simple mononucleotide microsatellites, accounting for 79.2% of all discovered SSR motifs (Figure 12). Among them, 37 are A/T mononucleotide repeats, and only one is a G/C repeat. There are eight dinucleotide repeats and two trinucleotide repeats. No simple sequence repeats with the core motifs of four nucleotides or longer were found. There are three microsatellites in the coding sequence of the gene, including two 10 bp A/T repeats, one located at the boundary of the coding sequence of the gene *sdh3*, and the other located in the exon 4 of the gene *nad1*, and a 12 bp AT repeat siting in the coding sequence of *mttB*. The type 1 mitogenome was dominant in the samples collected from Aminuis, and there was one interesting finding that the individual A11 had approximately 96% type 1 alleles, as well as 4% type 2 alleles at all differential loci between the two types of marama mitogenomes (Figure 13), which was also reflected in the 2k type 2 extra piece not shown in the figure. It has been confirmed that these minor alleles are not from homologous sequences in the nuclear genome or the chloroplast genome, suggesting the presence of two different mitogenomes in the same individual. It was previously reported that heteroplasmy also exists in the chloroplast genome of A11, with a minor genome frequency reaching 11% [42]. This means that in the chloroplasts and mitochondria of A11, both major and minor genomes exist, and the frequency of the minor genome is higher than that of the other studied plants. This is most likely to be caused by an accidental occurrence of paternal leakage. The proportions of mitochondrial and chloroplast minor genomes differed in A11, indicating that this may be true heterogeneity rather than due to the accidental mixing of samples. However, this is the only sample with high overall organelle genome heterogeneity, and more plants from Aminuis need to be sequenced and studied, especially from the vicinity of the collection of A11.

In the mitogenomes of other individuals, heteroplasmy was found to be less common than in their chloroplast genomes, and even low proportions of base substitutions below 2% were relatively rare, and even if present, many were found from the mitochondrial homologous segments in the nuclear genome. However, at a few differential loci, including the three consecutive substitutions at positions 127,684 to 127,686 and another substitution at 140,922 on chromosome LS1 (OK638188), obvious heteroplasmy could be seen in multiple individuals, with the proportion of minor alleles even up to 35%. The possible role, if any, of these loci in the evolution of individuals under environmental selection has also not been determined.

### 3.6. Sequence Transfer between Chloroplast and Mitochondrial Genomes

It was interesting to find that the chloroplast DNA insertions were concentrated in one of the subgenomic rings of the mitogenome (Figure 14). A low collinearity between the two genomes in these regions of similarity was seen, possibly due to each of the inserted cpDNA fragments being independently transferred. The largest of these fragments was 9798 bp containing the chloroplast pseudogenes *psbC*, *rps14*, *psaB*, and *psaA*, and the unchanged transfer RNA genes *trnG*, *trnM*, and *trnS*. A large number of variations were observed between the chloroplast and mitochondrial versions. Primers were designed to amplify across the two ends of this fragment from both the plastome and the mitogenome to verify its presence (Figure 15). On this long homologous DNA fragment, 20 loci showed differences between types, 13 of which were speculated to occur in the mitogenome and seven in the chloroplast genome (Table 7). In addition, the two organelle genomes differed at another 72 loci in this segment (this included 22 deletions, 6 insertions, and 44 SNPs), which are the same for both germplasms, making it difficult to tell where these variants arose. Ten variations were found in the gene sequences on this segment, of which only two synonymous substitutions occurred in the chloroplast genome, and the remaining eight were in the mitogenome, some of which may have a strong impact on transcription, including the introduction of early stop codons (Appendix A). However, only the sequences of the protein-coding genes were affected, while the sequences of the tRNA genes remained unchanged.

The mitogenome of *T. esculentum* was 399,572 bp (type 1) in length, and a total of 254 variations were found in the mitogenomes of the 84 individuals. The length of the chloroplast genome was 161,537 bp (type 1), and a total of 147 variations were found in the plastomes of these plants. The value of variation per nucleotide of the chloroplast genome is higher than that of the mitogenome. However, these chloroplast genes are originally conserved in the chloroplast genome, but after being inserted into the mitochondria, they become prone to mutation accumulation, which may be related to the loss of the original protection mechanism after entering the new environment, thus rendering them nonfunctional pseudogenes.

## 4. Discussion

The comparative analysis of the organelle genomes of 84 *T. esculentum* individuals revealed two germplasms with distinct mitochondrial and chloroplast genomes. The type 1 mitogenome contains two autonomous rings or five smaller subgenomic circular molecules with a total length of 399,572 bp. These two equimolar structures are thought to be interchangeable via recombination on three pairs of long direct repeats [31]. The type 2 mitogenome contains three circular molecules and one linear chromosome with a total length of 436,568 bp. The size of the assembled marama mitogenomes are close to that of some other legumes, 402.6 kb for *G. max*, 401.3 kb for *V. radiata*, 425.7 kb for *Pongamia pinnata*, and 380.9 kb for *L. japonicus* [59,85,86]. A mix of linear and circular molecules has also been previously proposed in the mitogenomes of plants such as *Lactuca sativa* and *S. tuberosum* [32,39]. Read alignment revealed a progressive decrease in coverage starting up to 18 kb from the chromosome ends, supporting the presence of linear molecules in the marama mitogenome. Both ends of the marama linear mitochondrial chromosome are long repetitive sequences, also present in other molecules, on which sequence invasion and recombination may occur to protect the linear chromosome from degradation [78]. The type 2 mitogenome also has a unique fragment of 2108 bp in length, likely derived from the same ancestral sequence as in other legumes such as *Lupinus* and *Indigofera*. This fragment was not found in any type 1 marama individuals and may be useful for the studies of marama germplasm typing and phylogeny. Since the phenotypic performances vary greatly among wild marama individuals, it is unclear whether these germplasmic genomic differences are associated with any phenotypes. In the future, it is expected that more samples will be collected, and genomic differences will be studied together with phenotypes to facilitate breeding efforts.

The structural variation resulted in an increased copy number of the genes *atp8*, *nad5* (exon3 and exon4), *nad9*, *rrnS*, *rrn5*, *trnC*, and *trnfM* in the type 2 mitogenome, but it is unknown whether this change is reflected in the gene expression level. The genes *nad1*, *nad4L*, *rps10*, and *mttB* were found to use alternative start codons ACG and ATT, similar to the mitogenome of the common bean [79]. Trans-splicing is considered to possibly play an important role in the activities of marama mitochondrial *nad* genes including *nad1*, *nad2*, and *nad5* because the exons of these genes were found to exist in different subgenomic structures, which is consistent with the previous study [89]. A total of 254 differential loci were found in the mitogenomes of the 84 *T. esculentum* individuals. Type 1 and type 2 mitogenomes differ from each other at 230 of these loci. However, the mitochondrial gene sequence is very conserved, in which only one of these 254 variations was found, which altered the amino acid sequence synthesized by *matR*, whose function is thought to involve the splicing of various group II Introns in Brassicaceae mitochondria [90].

The evolutionary study of the differential loci in the mitogenomes of *T. esculentum* found that the two types of plants fell into two clusters, as expected. The type 1 plants can be further divided into several clades; however, some of these geographic distribution-related differences may arise from sampling errors. For example, the sequencing of a larger number of samples confirmed that the widespread variation in three consecutive substitutions at positions 127,684 to 127,686 on chromosome LS1 (OK638188) may not be related to the geographic distribution of the plants. The phylogenetic tree constructed based on the mitochondrial genes of *T. esculentum* and other related Fabaceae species was consistent with the previously published one built on chloroplast genes [87]. The study found that *C. canadensis* and *T. esculentum* from the same subfamily Cercidoideae are closely related, and that they have more complete sets of mitochondrial genes than the Faboideae species, including *L. japonicus*, *M. sativa*, *M. pinnata*, *G. max*, and *V. radiata*. This is in line with previous studies where multiple gene losses occurred during the evolution of the Faboideae species [91].

Heteroplasmy in the mitochondrial genome of *T. esculentum* is not as prevalent as in its chloroplast genome, and most of the alternative alleles were actually found on the nuclear mitochondrial DNA segments [42]. In the mitogenome, heteroplasmy appears to be stably inherited, with very low levels, generally below 2%, but also with higher levels at certain loci, such as loci 127,684 to 127,686 on chromosome LS1. Among all samples, only one individual A11 from Aminuis had a generally high degree of heteroplasmy at most differential loci, consistent with what was seen in its chloroplast genome. Whether it is because of some of its own characteristics that A11 escaped the genetic bottleneck at the developmental stage is still unclear.

Chloroplast insertions are shown to be concentrated in one of the subgenomic rings of the mitogenome of *T. esculentum* with low collinearity. The mitogenomes of all studied marama individuals were found to contain a long chloroplast DNA insertion over 9 kb in length. Polymorphism studies of this fragment implied that the sequence in the chloroplast genome was protected by some mechanism, so only a small number of synonymous substitutions remained, but after being inserted into the mitogenome, a large number of mutations accumulated on it, progressively resulting in the loss of gene function, but the sequence of the tRNA gene has not changed and may still be functional [92,93,94].

## 5. Conclusions

The comparative analysis of 84 *T. esculentum* individuals revealed two distinct germplasms with significant differences in both the mitochondrial and chloroplast genomes. Unlike the previously known type 1 mitogenomes, the newly discovered type 2 mitogenomes consist of three circular molecules and one long linear chromosome. The type 2 mitogenomes also possess a unique 2108 bp fragment homologous to the Lupinus mitogenome. The structural variation increased the copy number of certain genes, including *nad5*, *nad9*, *rrnS*, *rrn5*, *trnC*, and *trnfM*, but the impact on their expression level remains uncertain. A total of 254 differential loci were identified in the populations, with 230 differences observed between the two germplasms. However, only one nonsynonymous mutation was found in the coding sequence of *matR*, indicating a conserved nature of plant mitochondrial genes. The evolutionary relationship between the mitogenome of marama and its related legume species is consistent with relationships built on chloroplast genes. Notably, both *C. canadensis* and *T. esculentum*, as species from the Cercidoideae subfamily, tend to possess a more complete set of mitochondrial genes compared to the Faboideae species, suggesting a loss of mitochondria genomic content during the evolution of the modern legume species. Heteroplasmy in the mitogenome of marama is not as prevalent as in its chloroplast genome, but it is concentrated at specific loci, including 127,684 to 127,686 on chromosome LS1, with the cause and effect remaining unclear. The mitogenome of marama contains a long chloroplast DNA insertion with a large number of polymorphisms. The study of this segment reveals the accumulation of mutations, leading to the loss of gene function, which possibly only occurs after the transfer into the new environment.

## Figures and Tables

**Figure 1 biology-12-01244-f001:**
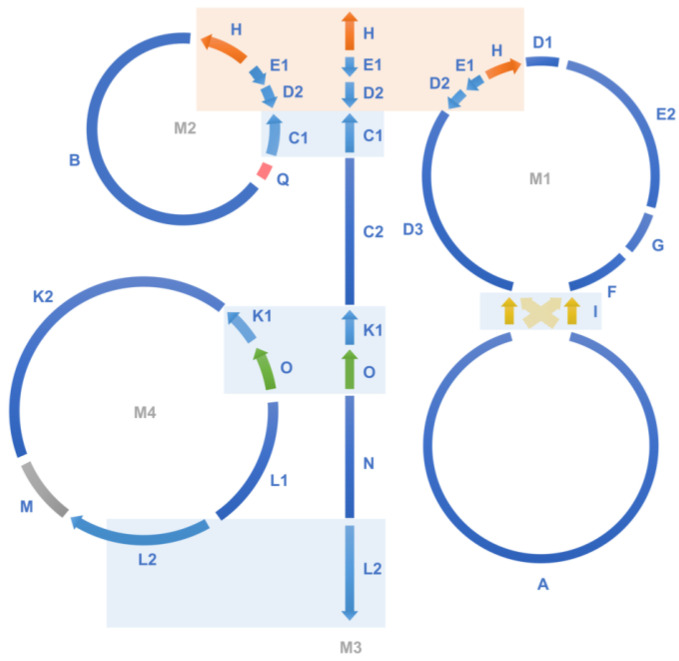
The assembly graph of the type 2 mitogenome of *T. esculentum*. The type 2 mitogenome consists of three circular molecules M1, M2, and M4, and one linear molecule M3, and they were built on 21 long scaffolds (Table 1). Blue blocks show double-copy regions that are identical between two chromosomes, and orange blocks indicate triple-copy regions owned by three chromosomes. Close sequencing coverage was found for single-copy regions of the four chromosomes. Recombination on a pair of long inverted repeats can change the junction of the upper and lower halves of M1 to that indicated by the yellow dashed arrows. The two structures before and after recombination have been confirmed via PacBio long reads, and their frequencies were close in the same individual (Appendix A). A long chloroplast insertion was found at the position of the gray segment M, about 9798 bp, and its length varied slightly among different individuals. This chloroplast insertion and the long repeats H, I, and O shown by colored arrows are also present in the type 1 mitogenome of *T. esculentum*. The type 1 mitogenome also has two rings, very similar to M1 and M4 here, but other molecules have undergone dramatic changes.

**Figure 2 biology-12-01244-f002:**
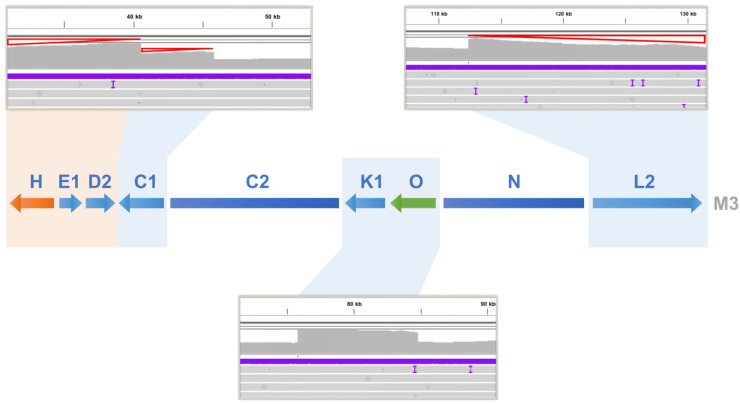
Changes in sequencing coverage on the type 2 mitogenome chromosome M3 of *T. esculentum*. PacBio HiFi reads from the type 2 individual Sample 4 were aligned to the multicopy regions of chromosome M3 using pbmm2. Chromosome M3 contains some long repeats that are identical to parts of other chromosomes, thus increasing the sequencing coverage of these regions in the alignment (Appendix A). At the positions of scaffolds C1, K1-O, and L2, with blue shading, the read depth was found to be doubled. At the location of scaffold H shaded in orange, the sequence depth was increased to 3-fold. However, a progressive decrease in coverage indicated by red triangles was also seen at both ends of M3, as some linear M3 chromosomes had degenerated at both ends without telomere protection. The colored arrows H and O represent long repetitive sequences also present in the type 1 mitochondrial genome of *T. esculentum*.

**Figure 3 biology-12-01244-f003:**
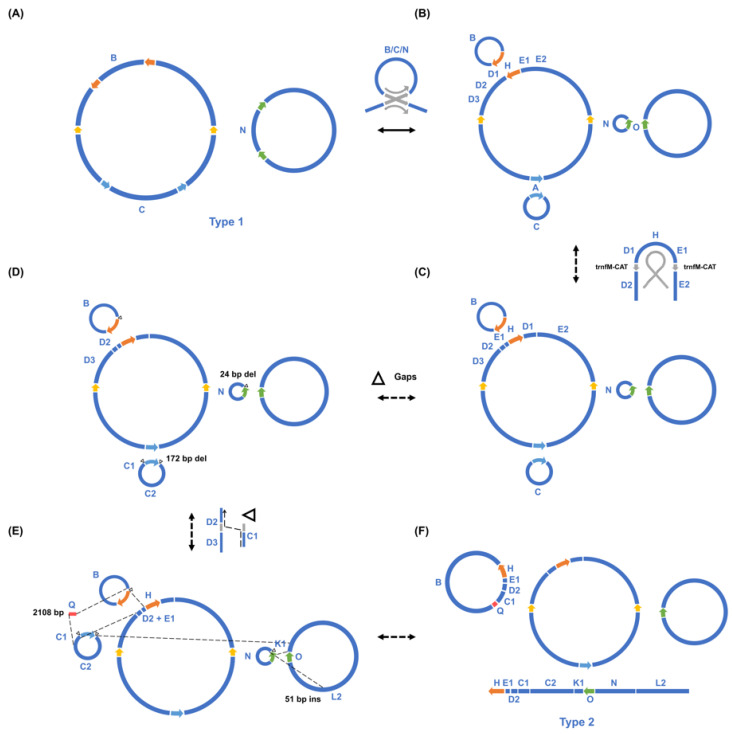
Step-by-step analysis of the structural differences between the two types of *T. esculentum* mitogenomes. The long repeats and some key components are represented by different colors. (**A**). The two autonomous circular chromosomes of type 1 *T. esculentum* mitogenome, LS1 (OK638188) (left) and LS2 (OK638189) (right). Colored arrows indicate the four pairs of long repeats (>1 kb). (**B**). Recombination on the direct repeats split the two large rings into five small circular molecules. Both conformations before and after recombination have been confirmed via PacBio reads to exist in type 1 individuals. (**C**). A rare recombination on a pair of *trnfM* genes at the junctions of D1 and D2, and E1 and E2, inverted the sequence D1-H-E1 in between. (**D**). Gaps with or without sequence deletions resulted in the three small circular chromosomes of the type 1 mitogenome present as different forms in type 2 individuals (Appendix A). (**E**). New DNA fragments, including a 2108 bp contig Q unique to type 2 individuals, joined originally remotely located sequences to form new structures. (**F**). The final type 2 mitogenome of *T. esculentum* with three circular chromosomes and one linear chromosome.

**Figure 4 biology-12-01244-f004:**
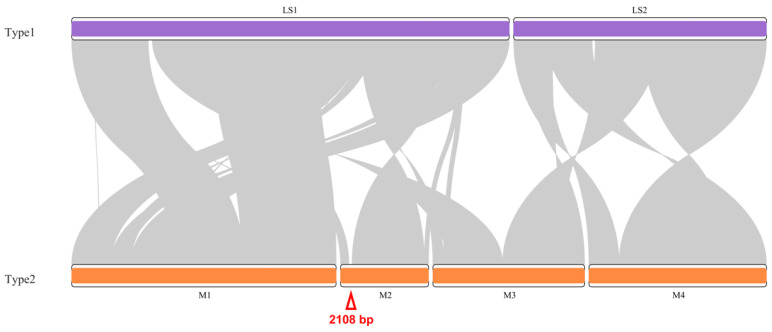
Synteny visualization of the two types of mitogenomes of *T. esculentum* via the R package RIdeogram after NUCmer alignment. The chromosomes of the type 1 mitogenome are shown in purple, and those of the type 2 mitogenome are shown in orange. The red triangle indicates the 2108 bp type 2 mitogenome exclusive fragment of *T. esculentum*.

**Figure 5 biology-12-01244-f005:**
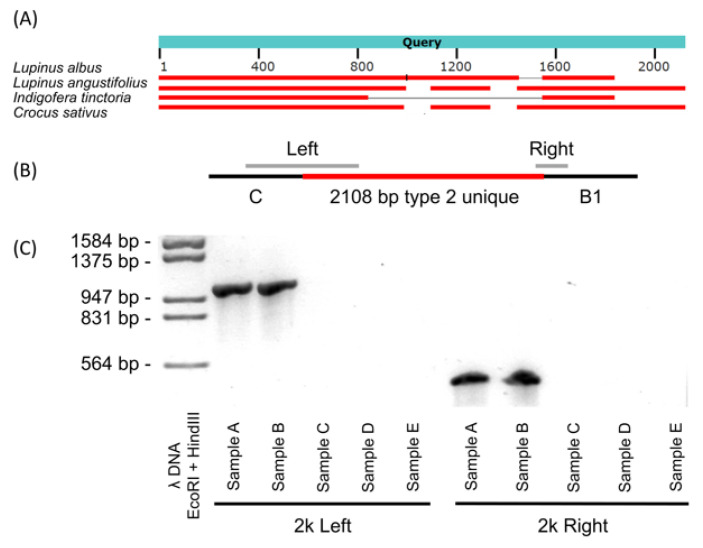
Homology analysis of the 2108 bp fragment unique to type 2 *T. esculentum* mitogenome and design of primers for its PCR identification. (**A**) The 2108 bp type 2 mitogenome exclusive sequence was blasted as a query in the NCBI database. Red horizontal bars indicate where database sequences are aligned, and separately aligned regions from the same database subject are connected by thin gray lines. (**B**) Two pairs of primers were designed to amplify products across both ends of the 2108 bp fragment (Appendix A). The estimated size of the left end product is 990 bp, and the right end product is 289 bp. (**C**) Gel image of PCR amplification of DNA from five randomly selected samples with the two pairs of primers was designed separately. The PCR products were electrophoresed on a 1.5% agarose gel at 110 V for 2 h.

**Figure 6 biology-12-01244-f006:**
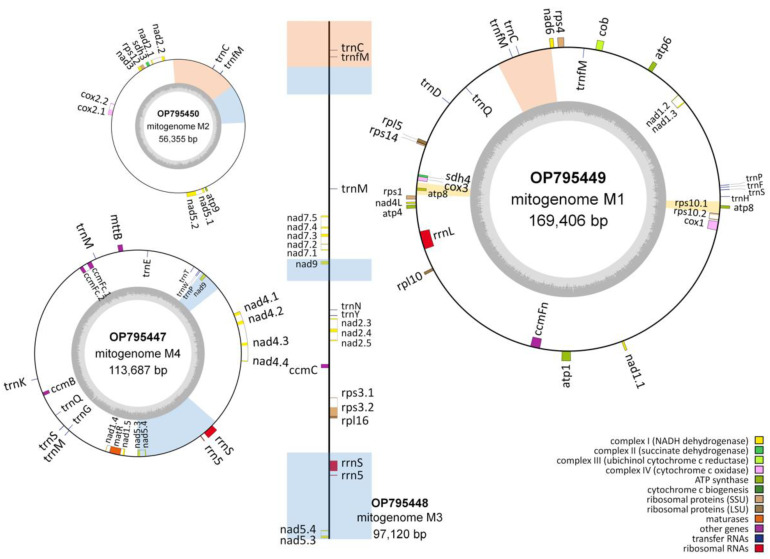
The map of type 2 *T. esculentum* mitogenome gene arrangement drawn via OGDRAW. The annotation was performed via MITOFY and BLAST on the mtDNA of individual Index1 from UP Farm and deposited in GenBank under accession numbers OP795447-OP795450. All type 2 plants were found to have a similar gene arrangement. The dark gray pattern in the inner circle indicates GC content. Genes are colored according to their function. The decimal part after the gene name indicates the order of the exons. Genes inside the circle are transcribed clockwise, while those outside the circle are transcribed counterclockwise for M1, and vice versa for M2 and M3. Blue blocks represent two-copy regions that are identical between two chromosomes, and orange blocks show three-copy regions that are the same across three chromosomes.

**Figure 7 biology-12-01244-f007:**
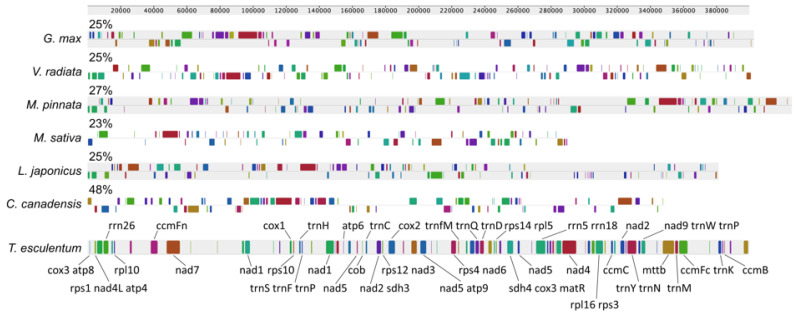
Synteny block diagram of the Mauve alignment between the mitogenomes of *T. esculentum* and six other Fabaceae species, *C. canadensis* (MN017226.1), *L. japonicus* (NC_016743.2), *M. sativa* (ON782580.1), *M. pinnata* (NC_016742.1), *G. max* (NC_020455.1), and *V. radiata* (NC_015121.1). Similarities (percentage of marama mitogenome covered) are labeled above the bars. Genes contained in the synteny blocks are marked via gene symbols.

**Figure 8 biology-12-01244-f008:**
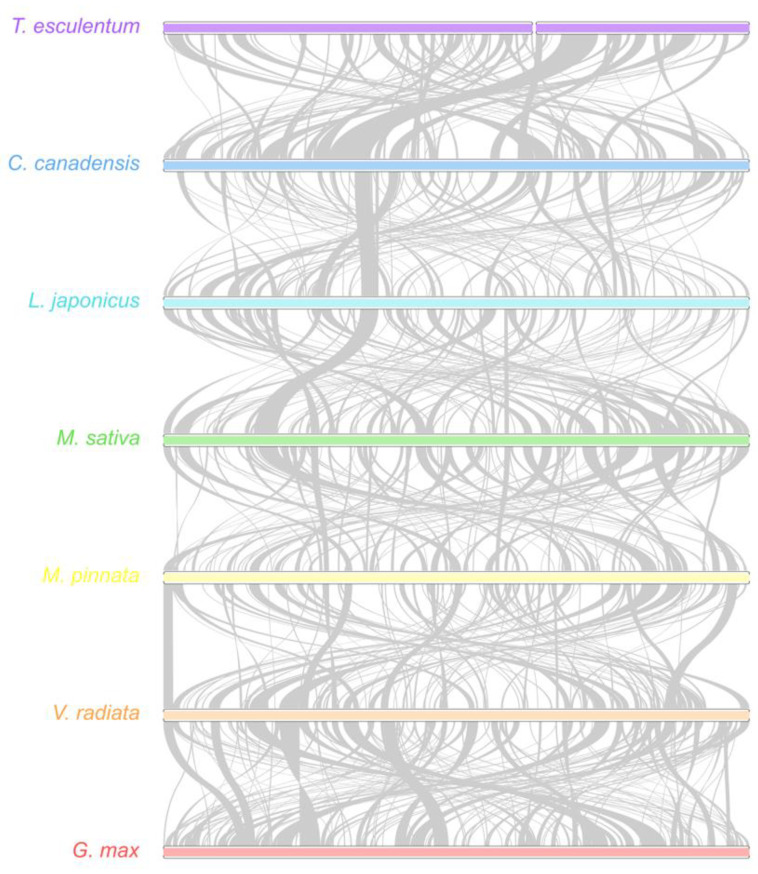
Synteny maps of the mitochondrial genomes of seven legume species. The colored bars represent the mt chromosomes of *C. canadensis* (MN017226.1), *L. japonicus* (NC_016743.2), *M. sativa* (ON782580.1), *M. pinnata* (NC_016742.1), *V. radiata* (NC_015121.1), *G. max* (NC_020455.1), and *T. esculentum*, which contains two chromosomes, LS1 (OK638188) and LS2 (OK638189). The gray ribbons indicate homologous sequences between the two neighboring species. The species were ordered according to the phylogenetic tree in Figure 9. Promer was used to detect syntenic regions between highly divergent genomes, which were then visualized via Rideogram package in R.

**Figure 9 biology-12-01244-f009:**
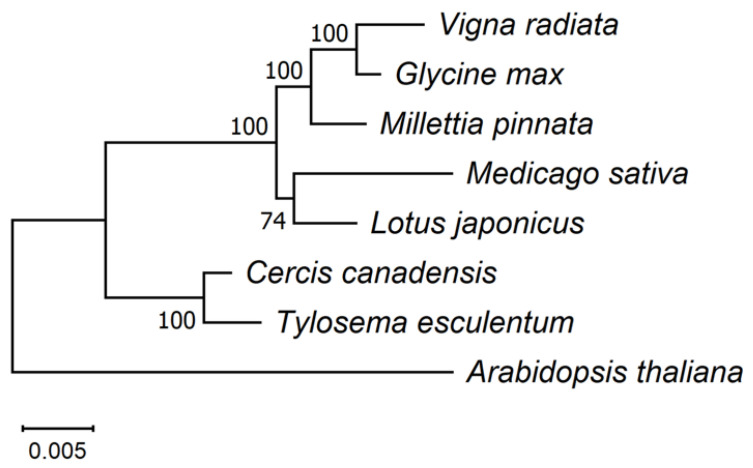
Maximum Likelihood (ML) phylogenetic tree with the Jukes–Cantor model based on artificial chromosomes concatenated by 24 conserved mitochondrial genes, *atp1*, *atp4*, *atp6*, *atp8*, *atp9*, *nad3*, *nad4*, *nad4L*, *nad6*, *nad7*, *nad9*, *mttB*, *matR*, *cox1*, *cox3*, *cob*, *ccmFn*, *ccmFc*, *ccmC*, *ccmB*, *rps3*, *rps4*, *rps12*, and *rpl16* from *A. thaliana* (NC_037304.1), *C. canadensis* (MN017226.1), *L. japonicus* (NC_016743.2), *M. sativa* (ON782580.1), *M. pinnata* (NC_016742.1), *G. max* (NC_020455.1), and *V. radiata* (NC_015121.1) in NCBI. The tree was drawn in Mega 11 after sequence alignment with Muscle v5. Percentage probabilities based on 1000 bootstrap replications are labeled on the branches. The topology was validated via the Bayesian inference phylogenetic tree drawn using BEAST (Appendix A).

**Figure 10 biology-12-01244-f010:**
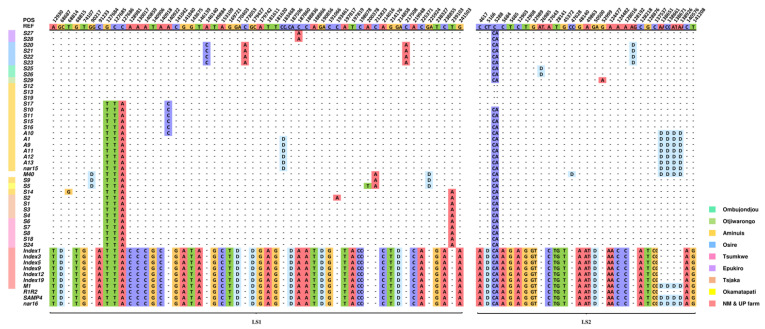
Nucleotide matrices showing the distribution of mitochondrial genome variations in the 43 independent individuals and 4 additional samples of unknown origin. The first row indicates the alleles of the type 1 reference mitogenome of *T. esculentum* (LS1:OK638188 and LS2: OK638189). From the second row onwards, only bases different from the reference are shown, and bases identical to the reference are replaced by dashes. The two types of mitogenomes also differ from each other at another 170 loci, not shown here to save space (no within-type differences were found at those 170 loci). The full variation distribution is shown in Appendix A. All insertions are represented by the first two bases and deletions by the letter “D”. The color bar to the left of the plant ID shows the source of the sample and is left blank for unknown sources.

**Figure 11 biology-12-01244-f011:**
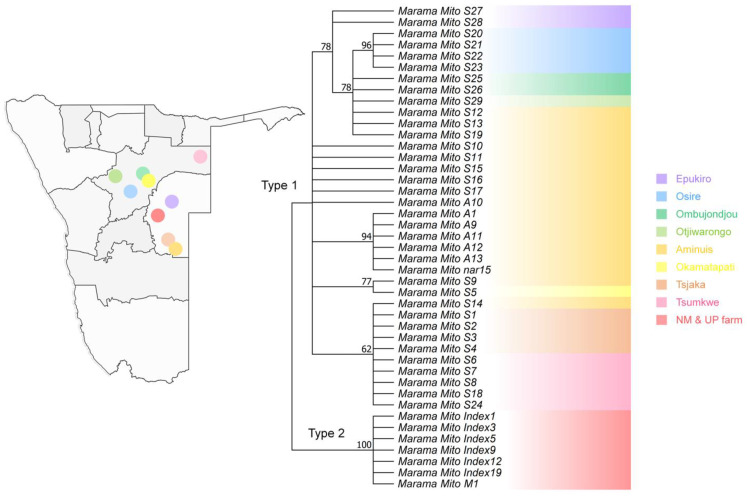
Maximum Likelihood (ML) phylogenetic tree with the Jukes–Cantor model built on artificial chromosomes concatenated via 40 bp fragments at each of the 254 differential loci in the mitogenomes of *T. esculentum* according to the mitogenome sequences of the 43 independent individuals. Frequencies from 1000 bootstrap replicates were labeled on the branches with 60% as cutoff. The topology was verified via the neighbor-joining method in Mega 11. Individuals with the same background color came from the same geographic location, and the sampling points were marked on the map of Namibia.

**Figure 12 biology-12-01244-f012:**
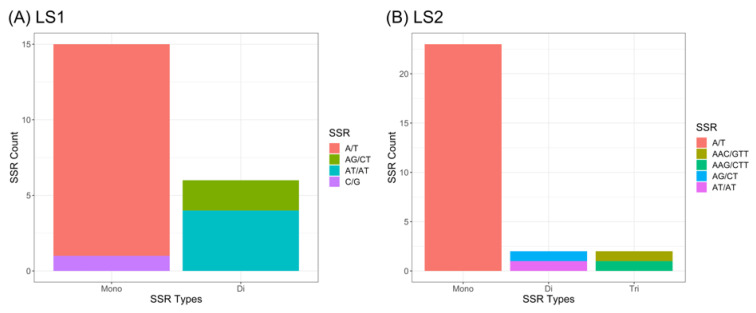
Distribution of SSR motifs of different repeat types in the type 1 reference mitogenome of *T. esculentum* analyzed via MISA. (**A**). Chromosome LS1 (OK638188). (**B**). Chromosome LS2 (OK638189).

**Figure 13 biology-12-01244-f013:**
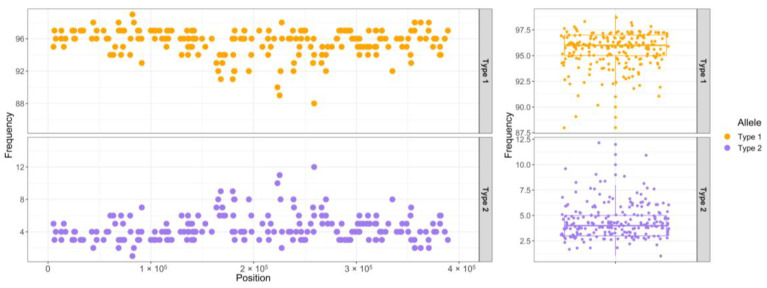
Allele frequency plot of all differential loci between the two types of mitogenomes of *T. esculentum* in Aminuis individual A11. The x-axis of the left panel indicates the position on the chromosome concatenated by the reference marama mitogenomes LS1 (OK638188) and LS2 (OK638189).

**Figure 14 biology-12-01244-f014:**
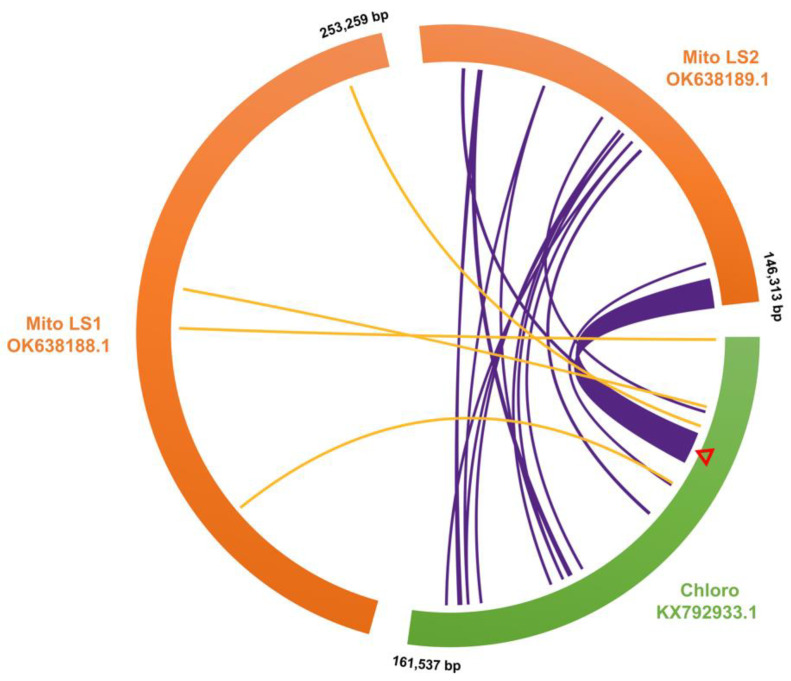
Map of chloroplast DNA insertions in the mitochondrial genome of *T. esculentum* drawn via TBtools Advanced Circos. The reference mitogenome chromosomes LS1 (OK638188) and LS2 (OK638189) were aligned with the chloroplast genome (KX792933) of *T. esculentum* using BLAST (Appendix A). The curves in the middle connect the homologous sequences of the plastome and mitogenome. The cpDNA insertions in the two mitochondrial chromosomes are colored orange and purple, respectively. cpDNA insertions were concentrated on the mitochondrial chromosome LS2, with only four short fragments on LS1. All the three chromosomes are circular but are shown here as linear, with numbers next to them indicating their length and orientation. The red triangle marks the position of the 9798 bp long homologous fragment.

**Figure 15 biology-12-01244-f015:**
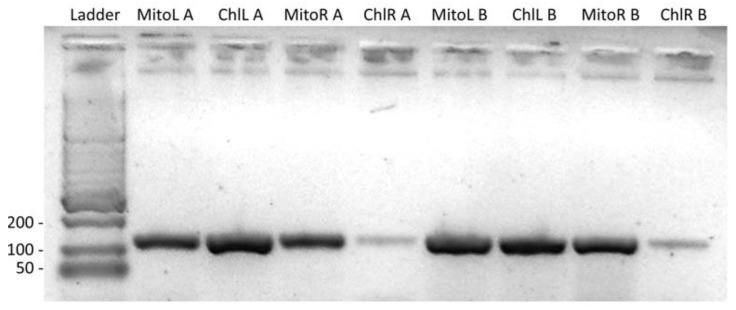
Amplification across the two ends of the 9798 bp homologous fragment of the mitochondrial and chloroplast genomes in two random samples A and B (Appendix A). Lane1, HyperLadder II; Lanes 2–5, DNA from sample A; Lanes 6–9, DNA from sample B. The products in lanes 2 and 6 were amplified with primers MitoLL and MitoLR, in lanes 3 and 7 were amplified with primers ChlLL and MitoLR, in lanes 4 and 8 were amplified with primers MitoRL and MitoRR, and in lanes 5 and 9 were amplified with primers MitoRL and ChlRR. The gel was run on a 1.5% agarose gel for 40 min at 100 V. The annealing temperature of 55 °C was a bit high for the primer ChlRR, resulting in low yields and faint bands in lanes 5 and 9. Decreasing the annealing temperature by 1 °C and repeating the PCR resulted in normal amplification.

**Table 1 biology-12-01244-t001:** Chromosome base composition of the type 2 *T. esculentum* mitogenome.

Molecule	A (%)	C (%)	G (%)	T (%)	G~C (%)	Length (bp)
M1	27.90	22.18	22.22	27.70	44.40	169,406
M2	27.60	21.63	23.26	27.52	44.90	56,355
M3	27.46	22.25	22.73	27.55	45.00	97,120
M4	27.70	22.58	22.44	27.29	45.00	113,687

**Table 2 biology-12-01244-t002:** Length of the primary scaffolds constituting the type 2 *T. esculentum* mitogenome.

Unit Number	Length (bp)	Unit Number	Length (bp)
A	82,874	K1	4052
B	39,273	K2	52,730
C1	30,017	L1	20,069
C2	5272	L2	22,301
D1	5866	N	27,581
D2	2722	O	4881
D3	26,671	M	9654
H	5212	Q	2108
E1	1768		
E2	25,384		
F	8590		
G	5798		
I	2265		

**Table 3 biology-12-01244-t003:** Gene annotation of the type 2 mitogenome of *T. esculentum*.

Category	M1	M2	M3	M4
Complex I (NADH dehydrogenase)	*nad1 #*, *nad4L*, *nad6*	*nad2 #*, *nad3*, *nad5 #*	*nad2 #*, *nad5 #*, *nad7 #*, *nad9*	*nad1 #*, *nad4 #*, *nad5 #*, *nad9*
Complex II (succinate dehydrogenase)	*sdh4*	*sdh3*		
Complex III (ubiquinol cytochrome-c reductase)	*cob*			
Complex IV (cytochrome-c oxidase)	*cox1*, *cox3*	*cox2 #*		
Complex V (ATP synthase)	*atp1*, *atp4*, *atp6*, *atp8 * (2)*	*atp9*		
Cytochrome c biogenesis	*ccmFn*		*ccmC*	*ccmB*, *ccmFc #*
Large subunit ribosomal proteins	*rpl5*, *rpl10*		*rpl16*	
Small subunit ribosomal proteins	*rps1*, *rps4*, *rps10 #*, *rps14*	*rps12*	*rps3#*	
Maturases				*matR*
Transport membrane protein				*mttB*
Ribosomal RNAs	*rrnL*		*rrn5*, *rrnS*	*rrn5*, *rrnS*
Transfer RNAs	*trnD-GTC*, *trnC-GCA*, *trnQ-TTG*, *trnH-GTG*, *trnfM-CAT * (2)*, *trnF-GAA*, *trnP-TGG*, *trnS-GCT*	*trnC-GCA*, *trnfM-CAT*	*trnN-GTT*, *trnC-GCA*, *trnfM-CAT*, *trnM-CAT*, *trnY-GTA*	*trnQ-TTG*, *trnE-TTC*, *trnG-GCC*, *trnK-TTT*, *trnM-CAT * (2)*, *trnP-TGG*, *trnS-TGA*, *trnT-TGT #*, *trnW-CCA*

Genes with introns are marked with #. Genes with multiple copy numbers on the same chromosome are labeled with *, and the numbers in parentheses indicate the corresponding copy numbers.

**Table 4 biology-12-01244-t004:** List of mitochondrial protein-coding genes lost during the evolution of some Fabaceae species (+ Present, - Absent).

Gene	*T. esculentum*	*C. canadensis*	*L. japonicus*	*M. sativa*	*M. pinnata*	*V. radiata*	*G. max*
*sdh3*	+	+	-	-	+	-	-
*sdh4*	+	+	-	-	-	-	-
*cox2*	+	+	+	+	+	-	+
*rpl2*	-	-	-	-	-	-	-
*rpl5*	+	+	+	+	+	+	-
*rpl10*	+	+	-	-	-	-	-
*rps1*	+	+	-	+	+	+	+
*rps2*	-	-	-	-	-	-	-
*rps7*	-	-	-	-	-	-	-
*rps8*	-	-	-	-	-	-	-
*rps11*	-	-	-	-	-	-	-
*rps13*	-	-	-	-	-	-	-
*rps19*	-	-	-	-	-	-	-

**Table 5 biology-12-01244-t005:** Total number of variations found when mapping the WGS reads of all 84 individuals to the type 1 *T. esculentum* reference mitochondrial genomes OK638188 and OK638189.

Chromosome	Variation Type	Type 2 vs. Ref.	Type 1 vs. Ref.	Total
LS1				
	Deletion	29 (29)	3 (3)	32
	Insertion	34 (34)	0	34
	SNP	79 (75)	13 (9)	88
LS2				
	Deletion	25 (21)	7 (3)	27
	Insertion	18 (17)	1 (0)	18
	SNP	54 (54)	1 (1)	55
Total		239 (230)	24 (16)	254

**Table 6 biology-12-01244-t006:** Variations found in *T. esculentum* mitochondrial gene sequences of the 84 individuals.

Position	Variant	Gene		Product
LS1 54113	Indel	*nad7*	Intron	NADH dehydrogenase subunit 7
LS2 2368	SNP	*matR*	Exon	maturase
LS2 35483	Indel	*nad4*	Intron	NADH dehydrogenase subunit 4
LS2 37645	SNP	*nad4*	Intron	NADH dehydrogenase subunit 4
LS2 39597	Indel	*nad4*	Intron	NADH dehydrogenase subunit 4
LS2 40927	SNP	*nad4*	Intron	NADH dehydrogenase subunit 4
LS2 41027	Indel	*nad4*	Intron	NADH dehydrogenase subunit 4
LS2 56879	SNP	*rps3*	Intron	ribosomal protein S3
LS2 69748	SNP	*nad2*	Intron	NADH dehydrogenase subunit 2
LS2 71388	Indel	*nad2*	Intron	NADH dehydrogenase subunit 2
LS2 71633	Indel	*nad2*	Intron	NADH dehydrogenase subunit 2

LS1 = OK638188, LS2 = OK638189.

**Table 7 biology-12-01244-t007:** Variations found in the 9798 bp homologous segment within the organelle genomes of the 84 *T. esculentum* individuals.

Position	Variation	Type 1 Chloro	Type 1 Mito	Type 2 Chloro	Type 2 Mito	Localization
35,570 *	SNP	Ref	Alt	Ref	Ref	Mito
36,347 *	DEL	Ref	Alt	Ref	Ref	Mito
36,942	SNP	Ref	Ref	Alt	Ref	Chloro
37,257	SNP	Ref	Alt	Alt	Alt	Chloro
37,813	SNP	Alt (some plants)	Ref	Ref	Ref	Chloro
38,753 *	DEL	Ref	Ref	Ref	Alt	Mito
38,975	SNP	Ref	Ref	Alt	Ref	Chloro
39,429 *	SNP	Ref	Alt	Ref	Ref	Mito
40,061 *	SNP	Ref	Alt	Ref	Ref	Mito
40,544 *	SNP	Ref	Alt	Ref	Ref	Mito
41,243 *	SNP	Ref	Alt	Alt	Alt	Chloro
41,716 *	SNP	Ref	Ref	Ref	Alt	Mito
41,718 *	SNP	Ref	Ref	Ref	Alt	Mito
42,587 *	SNP	Ref	Ref	Alt	Ref	Chloro
43,559	SNP	Ref	Ref	Ref	Alt	Mito
44,059	INS	Ref	Alt	Alt	Alt	Chloro
44,235	DEL	Ref	Alt	Ref	Ref	Mito
44,302	SNP	Ref	Alt	Ref	Ref	Mito
44,552	SNP	Ref	Ref	Ref	Alt	Mito
44,762	DEL	Ref	Ref	Ref	Alt	Mito

Ref refers to alleles identical to the type 1 chloroplast reference genome sequence. Alt refers to an allele that differs from the type 1 chloroplast reference genome sequence. Only two alleles, Ref or Alt, were seen at each locus. Comparing alleles in the same row, the locations where the mutations occurred can be estimated and written in the last column. This table does not include differences between the plastome and the mitogenome at another 72 loci on this fragment (no inter-type differences were seen at these loci). Positions in the gene sequence are marked with an asterisk *.

## Data Availability

The sequencing data used in this study are available and have been deposited under BioProject PRJNA897263 (https://www.ncbi.nlm.nih.gov/bioproject/897263).

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
