# Peer review of "Comparative Analysis of Tylosema esculentum Mitochondrial DNA Revealed Two Distinct Genome Structures"

_biology, 2023, doi:10.3390/biology12091244_

Round 1
Reviewer 1 Report
This study attempt to understand the genetic characteristics of Tylosema esculentum for promoting the selection of agronomically valuable germplasm. To address this question, this study assembled and compared the mitogenomes of 84 marama individuals, identifying variations in genome structure, single-nucleotide polymorphisms (SNPs), insertions/deletions (indels), heteroplasmy and horizontal transfer. Base on these variants, they identified two distinct germplasms with structural variants on many genes and exhibited differences at 230 loci. These results suggested that The evolutionary and comparative genomics analysis indicated that mitogenome divergence in marama might not be solely constrained by geographical factors.
1. Major comments:
a. Linear genome is a strong statement of the manuscript. PCR validation is required for the statement, as it is not difficult to valide it by PCR.
b. For confidence issue, at least some of the structural variations should be validated by PCR experiment
c. A discussion of the functional impact from mt genome divergence between marama bean and other species would be very helpful to our understanding of the evolution story.
d. The impact of gene transfer from chloroplast on the divergence of mt genome divergence between marama bean and other species, and on the divergence among marama beans, were also helpful for understanding the evolution history. I am really curious about whether the gene transfer event was the driver of the divergence.
2. Minor comments:
a. in line 254, different bootstrap cutoffs should be applied for showing the robustness of the tree inferred, as 40% is not a strict cutoff.
b. line288-290, lacking of figures or reference for supporting the described result
c. In line321, it is confusing that “reads found to stop in this range”. It would be great if the authors could clarify the describe.
d. Checking through the word use in the manuscript was recommended for ensuring the right information had been delivered.
Check through the MS and clarify some of the descriptions are required
Author Response
Response to reviewer 1
This study attempt to understand the genetic characteristics of Tylosema esculentum for promoting the selection of agronomically valuable germplasm. To address this question, this study assembled and compared the mitogenomes of 84 marama individuals, identifying variations in genome structure, single-nucleotide polymorphisms (SNPs), insertions/deletions (indels), heteroplasmy and horizontal transfer. Based on these variants, they identified two distinct germplasms with structural variants on many genes and exhibited differences at 230 loci. These results suggested that The evolutionary and comparative genomics analysis indicated that mitogenome divergence in marama might not be solely constrained by geographical factors.
- Major comments:
- Linear genome is a strong statement of the manuscript. PCR validation is required for the statement, as it is not difficult to validate it by PCR.
Our PacBio HiFi reads have an accuracy over 99.9%, an average length of over 10 kb, and coverage of 1300-1500 for single copy regions of the mitochondrial genome. These reads have been aligned to suspect positions to verify genome configuration. Some of these alignments have been included and shown in Supplementary Figures S13 to S18, with PacBio HiFi reads showing differences in the mitogenome structure of the two types of marama mitogenomes. In addition, a rough estimate of the ratio between the structures formed before and after recombination of these regions was also possible.
No PacBio reads indicated any continuing structure following the putative ends of the linear chromosome.
However, as the reviewers suggest, PCR amplification of long fragments spanning long repeats in the genome may help us better understand strand invasion and understand the proportion of structures before and after recombination, but amplified fragments longer than 20 kb are required to span long repeats in the mitogenome (most of our PacBio reads are shorter than this), which is very difficult to do even for PCR. Additionally, amplifying fragments within the putative linear chromosome would not resolve the structure and without an indication of where to place one of the PCR primers it is difficult to design amplifications that confirm the structure – one is looking for no amplification, a null PCR result which is difficult to interpret.
This linear fragment is not in a circular form as amplification with primers from each end of the chromosome do not result in amplified products.
- For confidence issue, at least some of the structural variations should be validated by PCR experiment
The primary structural variant resulting from the presence of the 2kb new fragment in the type 2 mitochondria was validated using PCR (Figure 5).
- A discussion of the functional impact from mt genome divergence between marama bean and other species would be very helpful to our understanding of the evolution story.
Horizontal transfer of genetic information is very common between the mitochondrial genome and other organelles in legumes. This has a significant impact on the genes contained and the overall size of the mitochondrial genome, also providing evolutionary guidance. However, the specific functions of these transfers remain unknown. Many genes, though lost in the mitochondrial genome, have been transferred to the nucleus where they may continue to function, and this has not been extensively studied. At this time, we do not have any information concerning the whole plant consequences of the mitochondrial variations and how those have influenced the evolution of this group of plants.
- The impact of gene transfers from chloroplast on the divergence of mt genome divergence between marama bean and other species, and on the divergence among marama beans, were also helpful for understanding the evolution history. I am really curious about whether the gene transfer event was the driver of the divergence.
The transfer of DNA sequences from chloroplasts to mitochondria and from mitochondria to nucleus is thought to have occurred very commonly and randomly during the evolution of angiosperms. As shown in Figure 14, the cpDNA insertions showed a very low collinearity supporting the speculation. The transferred genes from the chloroplast to the mitochondria were found to have accumulated mutations that result in the loss of their original function after transfer, as both the nucleic acid and protein sequences were drastically altered by the accumulated mutations. However, it is true that among legume species, some subfamilies retain a more complete set of mitochondrial genes, whereas other subfamilies may experience a significant loss of these genes. The regulatory mechanism behind these transfers is still unclear, and also their functional impact.
- Minor comments:
- in line 254, different bootstrap cutoffs should be applied for showing the robustness of the tree inferred, as 40% is not a strict cutoff.
Since the differences between marama mitochondrial genomes are very small, some of which are only one or two bases different, a cut off of 40% was initially used to show as many topological branches as possible, but in the analysis we only focused on relatively reliable clusters with high confidences. The figure has been redrawn to increase the cut off to 60%, however, this doesn’t change any of our conclusions.
Before
Figure 11. Maximum Likelihood (ML) phylogenetic tree with the Jukes-Cantor model built on artificial chromosomes concatenated by 40 bp fragments at each of the 254 differential loci in the mitogenomes of T. esculentum according to the mitogenome sequences of the 43 independent individuals. Frequencies from 1000 bootstrap replicates were labeled on the branches with 40% as cutoff. The topology was verified by the neighbor-joining method in Mega 11. Individuals with the same background color came from the same geographic location, and the sampling points were marked on the map of Namibia.
After
Figure 11. Maximum Likelihood (ML) phylogenetic tree with the Jukes-Cantor model built on artificial chromosomes concatenated by 40 bp fragments at each of the 254 differential loci in the mitogenomes of T. esculentum according to the mitogenome sequences of the 43 independent individuals. Frequencies from 1000 bootstrap replicates were labeled on the branches with 60% as cutoff. The topology was verified by the neighbor-joining method in Mega 11. Individuals with the same background color came from the same geographic location, and the sampling points were marked on the map of Namibia.
- line288-290, lacking of figures or reference for supporting the described result.
Figures S19 to S28 in the supplementary material support the classification of the mitochondrial genomes of these individuals whose chloroplast genome studies have been described in the previously published article (Li and Cullis, 2023). This is referenced in the paragraph.
Li, J., and Cullis, C. A. (2023). Comparative analysis of 84 chloroplast genomes of Tylosema esculentum reveals two distinct cytotypes. Frontiers in Plant Science, 13. https://doi.org/10.3389/fpls.2022.1025408
- In line321, it is confusing that “reads found to stop in this range”. It would be great if the authors could clarify the describe.
This has been rephrased as shown,
“In addition, in a very long range of 15 to 20 kb, the sequencing depth gradually decreased towards both ends with many PacBio reads reaching the ends within this range.”
Many PacBio long sequences stop at this range indicating a linear chromosome structure, and due to the lack of telomere protection, these mitochondria molecules may be degraded to exist at different lengths.
This has been explained in lines 321-326 as follows,
“This furthers confirms that this is a linear chromosome that exists in different lengths in cells because of the lack of telomere protection. Linear chromosomes have been found to stably exist in eukaryotic cells even in the absence of telomeres, through strand-invasion between terminal sequences and their homologous internal sequences to form t-loops to protect the chromosomes from degradation [78].”
- Checking through the word use in the manuscript was recommended for ensuring the right information had been delivered.
This has been done as suggested.
Reviewer 2 Report
Line 207 should be Go taq master mix (Promega, location) -consistent in the manuscript. Please do the same at other places in the manuscript like line 153. 275.
Figure 3 legend should be written correctly and the subsets numbers should be bold in the legends. The figure itself is not clear.
Author Response
- Line 207 should be Go taq master mix (Promega, location) -consistent in the manuscript. Please do the same at other places in the manuscript like line 153. 275.
This has been corrected at both locations as “GoTaq Master Mix (Promega, Madison, WI)”.
- Figure 3 legend should be written correctly and the subsets numbers should be bold in the legends. The figure itself is not clear.
Figure 3 has been redrawn and the font that was previously small and blurry has been enlarged. The title has been edited according to the requested format.
Before
Figure 3. Step-by-step analysis of the structural differences between the two types of T. esculentum mitogenomes. (i). The two autonomous circular chromosomes of type 1 T. esculentum mitogenome, LS1 (OK638188) (left) and LS2 (OK638189) (right). Colored arrows indicate the four pairs of long repeats (>1 kb). (ii). Recombination on the direct repeats split the two large rings into five small circular molecules. Both conformations before and after recombination have been confirmed by PacBio reads to exist in type 1 individuals. (iii). A rare recombination on a pair of trnfM genes at the junctions of D1 and D2, and E1 and E2, inverted the sequence D1-H-E1 in between. (iv). Gaps with or without sequence deletions resulted in the three small circular chromosomes of the type 1 mitogenome present as different forms in type 2 individuals (Figures S13-18). (v). New DNA fragments, including a 2,108 bp contig Q unique to type 2 individuals, joined originally remotely located sequences to form new structures. (vi). The final type 2 mitogenome of T. esculentum with three circular and one linear chromosomes.
After
Figure 3. Step-by-step analysis of the structural differences between the two types of T. esculentum mitogenomes. (A). The two autonomous circular chromosomes of type 1 T. esculentum mitogenome, LS1 (OK638188) (left) and LS2 (OK638189) (right). Colored arrows indicate the four pairs of long repeats (>1 kb). (B). Recombination on the direct repeats split the two large rings into five small circular molecules. Both conformations before and after recombination have been confirmed by PacBio reads to exist in type 1 individuals. (C). A rare recombination on a pair of trnfM genes at the junctions of D1 and D2, and E1 and E2, inverted the sequence D1-H-E1 in between. (D). Gaps with or without sequence deletions resulted in the three small circular chromosomes of the type 1 mitogenome present as different forms in type 2 individuals (Figures S13-18). (E). New DNA fragments, including a 2,108 bp contig Q unique to type 2 individuals, joined originally remotely located sequences to form new structures. (F). The final type 2 mitogenome of T. esculentum with three circular and one linear chromosomes.

Reviewer 3 Report
In the article by Li et al., "Comparative Analysis of Tylosema esculentum Mitochondrial DNA Revealed Two Distinct Genome Structures", they conducted a comparative analysis of 84 T. esculentum samples in order to understand their genetic characteristics. The authors have used PacBio and Illumina sequencing to assemble and compare the mitogenomes of 84 individuals from different locations in Namibia and South Africa. The study revealed distinct germplasms with significant differences in mitogenome structure and sequence. Comparative genomic analysis provided valuable insights into cytoplasmic genetic diversity and inheritance. The authors have done an excellent job designing, executing experiments, analyzing and presenting the data. The manuscript is well written and easy to understand.
Below, I provide comments to improve a few things in this article.
- Line 399: How many core mitochondrial genes and variable genes were annotated?
- Figure 5 (supplementary, page 27): The authors should run the gel longer for better ladder resolution and include an updated image.
- Please elaborate on the measure of heteroplasmy with proper literature references.
- Authors should summarize all the packages used for graphic representations in one place.
Author Response
In the article by Li et al., "Comparative Analysis of Tylosema esculentum Mitochondrial DNA Revealed Two Distinct Genome Structures", they conducted a comparative analysis of 84 T. esculentum samples in order to understand their genetic characteristics. The authors have used PacBio and Illumina sequencing to assemble and compare the mitogenomes of 84 individuals from different locations in Namibia and South Africa. The study revealed distinct germplasms with significant differences in mitogenome structure and sequence. Comparative genomic analysis provided valuable insights into cytoplasmic genetic diversity and inheritance. The authors have done an excellent job designing, executing experiments, analyzing and presenting the data. The manuscript is well written and easy to understand.
Below, I provide comments to improve a few things in this article.
- Line 399: How many core mitochondrial genes and variable genes were annotated?
This has been clarified as follows.
Before
Both type 1 and type 2 mitogenomes of T. esculentum were found to contain 35 unique protein-coding genes, 3 unique rRNA genes, and 16 different tRNA genes (Figure 6; Table 3)
After
Both types 1 and 2 mitochondrial genomes of T. esculentum share identical sets of genes, but differ in the copy numbers of certain genes. These genomes contain 35 unique protein-coding genes, 3 unique rRNA genes, and 16 distinct tRNA genes (Figure 6; Table 3).
- Figure 5 (supplementary, page 27): The authors should run the gel longer for better ladder resolution and include an updated image.
The gel has been re-run as shown in Figure S29.
Figure S29. Gel image of DNA PCR amplification of five randomly selected samples C2, 4, 32, 33, and 34 using two pairs of primers designed to span both ends of the 2k type 2 mitogenome unique fragment. Lane1, Lambda DNA/EcoRI + HindIII Marker; Lanes 2 and 7, sample C2; Lanes 3 and 8, sample 4; Lanes 4 and 9, sample 32; Lanes 5 and 10, sample 33; Lanes 6 and 11, sample 34; The products in lanes 2 to 6 were amplified with the 2k left forward and reverse primers, in lane 7 to 11 were amplified with the 2k right forward and reverse primers. The PCR products were electrophoresed on a 1.5% agarose gel at 110V for 2 hours.
The Figure 5 has been remade according to the new gel picture.
Before,
Figure 5. Homology analysis of the 2,108 bp fragment unique to type 2 T. esculentum mitogenome and design of primers for its PCR identification. (A) The 2,108 bp type 2 mitogenome exclusive sequence was blasted as a query in the NCBI database. Red horizontal bars indicate where database sequences are aligned, and separately aligned regions from the same database subject are connected by thin gray lines. (B) Two pairs of primers were designed to amplify products across both ends of the 2,108 bp fragment (Figure S29). The estimated size of the left end product is 990 bp, and the right end product is 289 bp. (C) Gel image of PCR amplification of DNA from six randomly selected samples with the two pairs of primers designed separately. The PCR products were electrophoresed on a 1.5% agarose gel at 80V for 1 hour.
After,
Figure 5. Homology analysis of the 2,108 bp fragment unique to type 2 T. esculentum mitogenome and design of primers for its PCR identification. (A) The 2,108 bp type 2 mitogenome exclusive sequence was blasted as a query in the NCBI database. Red horizontal bars indicate where database sequences are aligned, and separately aligned regions from the same database subject are connected by thin gray lines. (B) Two pairs of primers were designed to amplify products across both ends of the 2,108 bp fragment (Figure S29). The estimated size of the left end product is 990 bp, and the right end product is 289 bp. (C) Gel image of PCR amplification of DNA from five randomly selected samples with the two pairs of primers designed separately. The PCR products were electrophoresed on a 1.5% agarose gel at 110V for 2 hours.
- Please elaborate on the measure of heteroplasmy with proper literature references.
The criteria were established by ourselves. Previously, the marama chloroplast genome paper followed the same standards and provided more detailed explanations, which have been cited here. Different studies have adopted different allele frequency cutoffs, such as 0.5% (Carbonell-Caballero et al., 2015), 2% (Wallace and Chalkia, 2013), etc., which largely depend on the depth of sequencing data and the purpose of the experiment. The rest of the conditions are designed to remove interference from low-quality sequencing bases and strand bias., as mentioned in many other articles (Guo et al., 2014; Fazzini et al., 2021).
Wallace, D. C., and Chalkia, D. (2013). Mitochondrial DNA genetics and the Heteroplasmy Conundrum in evolution and disease. Cold Spring Harbor Perspectives in Biology, 5(11). https://doi.org/10.1101/cshperspect.a021220
Carbonell-Caballero, J., Alonso, R., Ibañez, V., Terol, J., Talon, M., and Dopazo, J. (2015). A phylogenetic analysis of 34 chloroplast genomes elucidates the relationships between wild and domestic species within the genus citrus. Molecular Biology and Evolution, 32(8), 2015-2035. https://doi.org/10.1093/molbev/msv082
Guo, Y., Li, C. I., Sheng, Q., Winther, J. F., Cai, Q., Boice, J. D., and Shyr, Y. (2013). Very Low-Level heteroplasmy MTDNA variations are inherited in humans. Journal of Genetics and Genomics, 40(12), 607–615. https://doi.org/10.1016/j.jgg.2013.10.003
Fazzini, F., Fendt, L., Schönherr, S., Forer, L., Schöpf, B., Streiter, G., Losso, J. L., Kloss-Brandstätter, A., Kronenberg, F., and Weissensteiner, H. (2021). Analyzing Low-Level mtDNA Heteroplasmy—Pitfalls and Challenges from Bench to Benchmarking. International Journal of Molecular Sciences, 22(2), 935. https://doi.org/10.3390/ijms22020935
- Authors should summarize all the packages used for graphic representations in one place.
All data visualization methods are covered in each subsection of the material, not sure if we need to repeat them again in a separate paragraph.
